

# Leaching of inorganic and organic phosphorus and nitrogen in contrasting beech forest soils – seasonal patterns and effects of fertilization

Jasmin Fetzer[1,2], Emmanuel Frossard[2], Klaus Kaiser[3], Frank Hagedorn[1]

[1]: Forest Soils and Biogeochemistry, Swiss Federal Institute for Forest, Snow and Landscape Research WSL, Birmensdorf, Switzerland
[2]: Department of Environmental Systems Science, ETH Zurich, Zürich, Switzerland
[3]: Soil Science and Soil Protection, Martin Luther University Halle-Wittenberg, Halle (Saale), Germany

*Correspondence to*: Jasmin Fetzer (jasmin.fetzer@wsl.ch)

**Abstract.** Leaching is one major pathway of phosphorus (P) and nitrogen (N) losses from forest ecosystems. Using a full factorial N×P fertilization and irrigation experiment, we investigated the leaching of dissolved organic and inorganic P (DOP and DIP) and N (DON and DIN) from organic layers (litter, Oe/Oa horizons) and mineral A horizons at two European beech sites of contrasting P status. Leachates showed highest DIP and DIN concentrations in summer and lowest in winter, while

dissolved organic forms remained rather constant throughout seasons. During the dry and hot summer 2018, DOC:DOP and DOC:DON ratios in leachates were particularly narrow, suggesting a release of microbial P due to cell lysis by drying and rewetting. This effect was stronger at the low-P site. The estimated annual mean fluxes from the Oe/Oa horizons in the non-fertilized treatment were 60 and 30 mg m$^{-2}$ yr$^{-1}$ for total dissolved P and 730 and 650 mg m$^{-2}$ yr$^{-1}$ for total dissolved N at the high-P and the low-P site, respectively. Fluxes of P were highest in the organic layers and decreased towards the A horizon,

likely due to sorption by minerals. Fertilization effects were additive at the high-P, but antagonistic at the low-P site: At the high-P site, fertilization with +N, +P, and +N+P increased total P fluxes from the Oe/Oa horizon by +33, +51, and +75%, while the respective increases were +198, +156, and +10% at the low-P site. The positive N-effect on DIP leaching possibly results from a removed N limitation of phosphatase activity at the low-P site. Fluxes of DOP remained unaffected by fertilization. Fluxes of DIN and DON from the Oe/Oa horizons increased upon +N and +N+P, but not upon +P fertilization.

In conclusion, the estimated P fluxes from the A horizons were comparable in magnitude to reported atmospheric P inputs, suggesting that these systems do not deplete in P due to leaching. However, a particularly high sensitivity of DIP leaching to hotter and drier conditions suggests accelerated P losses under the expected more extreme future climate conditions. Increases of P leaching due to fertilization and drying-rewetting were higher in the low-P system, implying that the low-P system is more susceptible to environmental future changes.



## 1 Introduction

Leaching is one major pathway of phosphorus (P) and nitrogen (N) loss from forest ecosystems (e.g. Bol et al., 2016; Hedin et al., 1995). Leaching refers to water flow related transport of solutes within and from soil. Leaching losses are a result of the balance between mineralization, dissolution, and desorption processes on one side, and biological uptake as well as abiotic

precipitation and sorption on the other side. High water flow conditions may support leaching since transport becomes faster than the uptake of nutrients by roots and microorganisms and sorption (e.g. Backnäs et al., 2012; Barrow, 1983). Phosphorus and N are leached in dissolved inorganic and organic forms, or sorbed to colloids, with concentrations and fluxes varying strongly among soils and ecosystem types (Bol et al., 2016; Kaiser et al., 2003; Qualls and Haines, 1991a). Dissolved organic N was found to dominate the N losses from unpolluted ecosystems (Perakis and Hedin, 2001, 2002), while inorganic N

dominates in leachates from most temperate forest ecosystems receiving high atmospheric N deposition (Hagedorn et al., 2001). Substantially less is known about P leaching (Bol et al., 2016; Hannapel et al., 1964a, 1964b), but the importance of organic forms for leaching may be even greater for P than for N (Qualls, 2000; Qualls and Haines, 1991a). Inorganic P forms strongly bind to or become incorporated into secondary minerals (e.g. Walker and Syers, 1976), or can be taken up by roots and microorganisms. Thus, concentrations of inorganic P in soil solutions are usually low even in soils where high amounts of

soil organic matter are mineralized (Bol et al., 2016; Kaiser et al., 2000; Qualls and Haines, 1991a). Some organic P compounds sorb less strongly on mineral surfaces than inorganic P (Anderson and Magdoff, 2005; Berg and Joern, 2006; Celi et al., 2003; Lilienfein et al., 2004), and it has been suggested that the most P-rich fractions of dissolved organic matter (DOM) are more mobile than the rest (Frossard et al., 1989; Kaiser et al., 2000, 2001; Qualls, 2000; Qualls and Haines, 1991a). Accordingly, much of the P in soil solution has been found in organic form (Hedin et al., 2003; Kaiser et al., 2000, 2003; Qualls, 2000;

Qualls and Haines, 1991b), and might lead to long-term P depletion of soils (Alvarez-Cobelas et al., 2009; Frossard et al., 1989; Hedin et al., 2003).

As outlined above, leaching is the net result of the interplay of biotic and abiotic processes: Models of C-N-P cycles assume that microbial activity is driven by temperature, moisture, and organic matter bioavailability, being key determinants for nutrient mineralization in soil (Colman and Schimel, 2013; Davies et al., 2016; Yu et al., 2020). Besides seasonal variation

(Kaiser et al., 2003; Kalbitz et al., 2000; Michalzik et al., 2001), extreme changes in temperatures by freeze-thaw cycles or in the precipitation regime were found to increase nutrient release (e.g. Gao et al., 2020, 2021). Drying-rewetting effects can trigger release pulses of nutrients (Birch, 1958) that are then prone to leaching (Achat et al., 2012; Blackwell et al., 2010; Borken and Matzner, 2009; Brödlin et al., 2019b; Dinh et al., 2017, 2016). Climatic warming and the increasing frequency and severity of droughts influence soil organic matter (SOM) dynamics and the associated release and leaching of P and N

(e.g. Gao et al., 2020).

In addition, the nutritional status of soils, defined by parent material, climatic conditions, and atmospheric input (Augusto et al., 2017), exerts strong influence on nutrient release (Mooshammer et al., 2014). The C:N:P stoichiometry of SOM has been identified as a key parameter: While critical C-to-nutrient ratios – above which nutrients that are mineralized during



decomposition become immobilized by soil microorganisms and below which they are released in excess of biological demand
– are well established for N, but remain uncertain for P (e.g. Davies et al., 2016; Mooshammer et al., 2014). However, the concept of resource limitation has shifted from an earlier paradigm of single-resource limitation (e.g. van der Ploeg et al., 1999) towards co-limitation by multiple resources (e.g. Harpole et al., 2011). Synergistic interactions between N and P have frequently been observed in aquatic and terrestrial ecosystems (Elser et al., 2007), but have - to our knowledge - not been studied in field studies on nutrient leaching in temperate forest ecosystems. Rising atmospheric $CO_2$, N, and P depositions may
induce imbalances between C, N, and P, impacting the C:N:P ratio of SOM and hence, the cycling and leaching of P and N (Fröberg et al., 2013; Mortensen et al., 1998; Peñuelas et al., 2013; Vogel et al., 2021; Wang et al., 2014).

While several studies addressed these potential biotic and abiotic factors individually, none has examined simultaneous effects under field conditions. Here, we studied the leaching of dissolved organic and inorganic P and N in temperate beech forest soils as affected by seasons, nutrient status, and fertilization. For that, we used zero-tension lysimeters in three soil horizons
that were artificially irrigated to standardize water flow. To cover different nutrient status, we took advantage of two sites of contrasting nutrient availability (a low-P sandy soil with low sorption capacities and a high-P loamy soil on basalt with high sorption capacities) that were subjected to a full factorial N×P fertilization experiment. To cover seasonal differences, we sampled leachates and soil solutions five times during 18 months.

With this study we aimed to quantify annual organic and inorganic P and N fluxes from organic layers and from the mineral
topsoil. We primarily focused on P fluxes since the leaching of N has been extensively studied before. We hypothesized: (i) Leaching of DIP and DIN from organic forest floor layers will show stronger seasonal variations and stronger fertilization effects than DOP and DON, as inorganic forms are more strongly controlled by mineralization and biotic uptake; in the mineral soil, seasonal and fertilization effects on leaching will be superimposed by sorption processes. (ii) The contribution of dissolved organic forms to total P and N leaching from organic layers will be greater at the low-P site due to stronger biotic uptake of
inorganic nutrients. (iii) Fertilization with N and P removes nutrient limitations, and thus, will enhance P and N leaching, with effect sizes depending on the site, due to differences in nutrient status and mineral assemblage; effect sizes will also differ between P and N, due to differences in sorption and uptake affinities of the two nutrients. (iv) We expect synergistic effects of the combined N and P fertilization since N addition will remove N limitation for biologically mediated P mobilization processes.

## 2 Material and methods

### 2.1 Study site description

The study was conducted in two mature beech forest stands in Germany with contrasting parent material and P availability. The soil with high P stock (Table 4) is a loamy Cambisol developed on basalt at Bad Brückenau (BBR, 809 m a.s.l., 50.35° N, 9.27° E, referred to "high-P site"). The soil with the lower P stock is a sandy Cambisol, featuring a thicker organic layer and
initial podzolization at Unterlüss (LUE, 115 m a.s.l., 52.8° N, 10.3° E, referred to "low-P site") that developed from glacial





till. The organic and A horizons of the sites have greater C:P ratios at the low-P site, but similar C:N ratios (Table 1, SI 1 and SI 8). The differing P status is also mirrored by leaf P, being higher at the high-P site with $0.71\pm0.02$ g P $kg_{leaves}^{-1}$ compared to $0.58\pm0.2$ g P $kg_{leaves}^{-1}$ at the low-P site. Compared to the low-P site, the high-P site is characterized by a slightly higher pH (3.2 vs. 3.0), higher cation exchange capacity (371 vs. 108 meq $kg^{-1}$) and higher proportion of aluminum and iron oxides (29.3 and 8.4 vs. 0.9 and 0.3 g $kg^{-1}$ in the A horizon, Lang et al., (2017), SI 1). More details on the two sites are reported in Lang et al. (2017).

**Table 1: Soil organic carbon (SOC), total soil nitrogen (N), and total soil phosphorus (P) concentrations in litter, Oe/Oa, and A horizons from each the control treatment (unfertilized) at the high-P and the low-P site. Samples were taken in July 2019; values represent means ± standard error of three field replicates.**

| Site | Horizon | Horizon thickness[+] | SOC | Total soil N | Total soil P | Resin-extractable P* | Total organic P fraction |
|---|---|---|---|---|---|---|---|
| | | cm | g C $kg^{-1}$ | g N $kg^{-1}$ | g P $kg^{-1}$ | g P $kg^{-1}$ | % of total P** |
| **high-P site** | Litter | 2 | 451 (± 1) | 17 (± 0) | 0.96 (± 0.08) | NA | 67 (± 7) |
| **high-P site** | Oe/Oa | 2.5 | 352 (± 17) | 19 (± 0.3) | 2.3 (± 0.32) | 0.145 (± 0.02) | 81 (± 2) |
| **high-P site** | A | 5 | 178 (± 20) | 12 (± 1.3) | 3.02 (± 0.28) | 0.039 (± 0.01) | 85 (± 3) |
| **low-P site** | Litter | 4.5 | 391 (± 9) | 14 (± 0.3) | 0.79 (± 0.08) | NA | 70 (± 1) |
| **low-P site** | Oe/Oa | 6 | 230 (± 25) | 11 (± 1.5) | 0.53 (± 0.08) | 0.095 (± 0.01) | 81 (± 4) |
| **low-P site** | A | 5 | 70 (± 14) | 3.4 (± 0.7) | 0.17 (± 0.02) | 0.005 (± 0) | 85 (± 2) |

[+] average horizon thickness above installed lysimeters (n = 12)

* measured by M. Siegenthaler on air-dried material, sampled in September 2018 from the same sites

** total organic P determined after Saunders and Williams (1955) and related to total P determined in $H_2SO_4$ extracts

## 2.2 Experimental set-up and in situ measurements

### 2.2.1 Fertilization

At each site, plots of 20 m × 20 m were established with 20 m space in between. The plots were fertilized with either +N, +P, or +N+P, plus unfertilized control plots. They were replicated three times and arranged in blocks, resulting in a total of 12 plots per site (details on experimental set-up are given in e.g. Hauenstein et al., 2020). 5.0 g P $m^{-2}$ were applied as $KH_2PO_4$ in a single dose in 2016 (0.6 and 3.0% of total P stock at the high-P and the low-P site, Table 5). This amount accounts for 0.6% and 3% of the total P stock within the top 1 m of soil plus forest floor at both sites (Table 5). To compensate for the K input to the P fertilized plots, 6.3 g $m^{-2}$ were applied as KCl at the N fertilized and control plots. Nitrogen was added as $NH_4NO_3$ in five equal doses from 2016 to 2018, amounting to a total of 15 g N $m^{-2}$ (1.2 and 2.1% of total N stock at the high-P and the low-P site, Table 6).



### 2.2.2 Zero-tension lysimeters

Zero-tension lysimeters were installed beneath the litter layer, the fermented/humified horizon (Oe/Oa), and the mineral topsoil (A) in November 2017 at both study sites. Zero-tension (or gravity) lysimeters mainly sample macro-pore flow during rain events and when the soil is draining to field capacity (Litaor, 1988) and, therefore, provide a sample of soil solution exported

from the respective soil horizon. The two uppermost lysimeters were 20 cm × 20 cm acrylic glass plates with a mesh and three holes ensuring contact with the soil underneath. The third lysimeter in the mineral A horizon was a 19.5 cm × 25.5 cm pod with a 3.3 cm high rim that was filled with three layers of acid-washed quartz sand of different grain sizes to ensure a hydrological continuum with the mineral soil. The lysimeters at the three depths were installed adjacent to each other. They were slightly inclined and connected to polyethylene (PE) bottles. Following installation, the plots were left to recover from

disturbance for five months. For more details see Fetzer et al. (2021) and SI 2.

### 2.2.3 Artificial irrigation and sampling

To measure solute leaching under standardized conditions with little storage time, an area of 1 $m^2$ above the installed lysimeters at each plot was irrigated in April 2018, July 2018, October 2018, February/March 2019, and in July 2019. The irrigation water was P- and N-free artificial rainwater matching pH (5.5) and electrical conductivity (25 µS $cm^{-1}$) of the average local

throughfall of both sites. The artificial rainwater was applied at a constant rate of 20 L $h^{-1}$ $m^{-2}$ using an Accu-Power sprayer (Birchmeier Sprühtechnik AG, Switzerland). This volume corresponds to 71% and 61% of the pore volume above the lysimeter in the mineral soil at the low-P and the high-P site, respectively. Soil solutions were collected over a period of 1.5 hours from the start of the irrigation. The collection represents the "first flush" leaching (Bol et al., 2016; Makowski et al., 2020; Rinderer et al., 2021). The application rate represents maximum rainfall intensities at the study sites. Rainfall intensities larger than 20

L $h^{-1}$ $m^{-2}$ have been observed once at the low-P site and three times at the high-P site during the last 10 years (Bayerische Landesanstalt für Wald und Forstwirtschaft (LWF) and Nordwestdeutsche Forstliche Versuchsanstalt (NW-FVA)). The three irrigations in 2018, in total 60 L $m^{-2}$, accounted for approx. 8% of measured throughfall at the high-P site and 16% at the low-P site (cf. Table 2). The two irrigations in 2019 added 40 L $m^{-2}$. Overall, leachates were collected at 5 dates, at 2 sites, at 3 soil depths, and at 4 fertilization treatments applied in three blocks, resulting in 360 samples. Leachates were collected in PE

bottles, stored in cooling boxes, and transported within 24 to 48 hours to the laboratory. Samples were filtered through 0.45 µm nitrocellulose filters (GVS Life Sciences, Zola Predosa, Italy), and stored at 4°C prior to analysis.

To quantify annual natural water and solute flux, we additionally collected leachate draining into the lysimeters between samplings, using polyethylene canisters. To keep evaporation minimal, the sampling containers were placed in a covered soil pit. Their water volume was determined gravimetrically.






### 2.3 Leachate analysis

Dissolved inorganic P was estimated spectrophotometrically as molybdate-reactive P (MRP), using the molybdate-ascorbic acid method (Murphy and Riley, 1962), and a flow injection analyzer (Scan+, Skalar, Breda, The Netherlands). Dissolved total P (DTP) concentrations in leachates were measured by inductive coupled plasma-optical emission spectroscopy (ICP-OES, Ultima 2, Horiba Jobin-Yvon, Longjumeau, France). Dissolved organic P was calculated as the difference between TDP and DIP. Concentrations of dissolved organic carbon (DOC) and total nitrogen (TN) were measured with a FormacsHT/TN analyzer (Skalar). Dissolved nitrate concentrations were measured by ion chromatography (ICS 3000, Dionex, Sunnyvale, CA, USA) and dissolved ammonium concentrations with a FIAS-300 (Perkin-Elmer, Waltham, MA, USA). Electrical conductivity was measured with a LF 325 probe (WTW, Weilheim, Germany) and pH with a LL ecotrode (Metrohm, Herisau, Switzerland).

### 2.4 Soil samples

Soil samples of each plot and horizon were taken in July 2019 (2 sites × 12 plots × 3 horizons, total n = 72). The material was freeze-dried and ground with a ball mill. The concentrations of N and C were measured using an automated elemental analyzer (Euro EA3000: Euro Vector, Pavia, Italy). Fused beads of sample aliquots ashed at 1000°C were used to determine total P by sequential wave-length dispersive X-ray fluorescence spectroscopy (S8 Tiger Series 2, Bruker AXS, Karlsruhe, Germany). Phosphorus extraction with resin ($P_{Resin}$) (Hedley et al., 1982; Moir and Tiessen, 2007) was carried out using samples taken in autumn 2018 at 1:30 soil:solution ratio, followed by colorimetric phosphate determination (UV-1800, Shimadzu, Canby, USA) with the malachite green method of Ohno and Zibilske (1991). Total organic P and $H_2SO_4$ extractable total P were determined with the ignition method (Saunders and Williams, 1955). Two subsamples of 0.5 g of dried and < 2 mm-sieved soil samples were taken and one of them being ignited (550°C, 2 h). Ignited and unignited soils were extracted with 25 mL of 0.5 M $H_2SO_4$ for 16 h and subsequently filtered. In both extracts inorganic P was determined with the Malachite green method (Ohno and Zibilske, 1991). Organic P was calculated as the difference between the inorganic P in the ignited and unignited samples.

### 2.5 Soil microclimatic measurements

Soil temperature and soil moisture sensors were installed in April 2018 and remained in place throughout the experimental duration. Soil temperature (5 cm depth) was recorded by buried iButtons (iButton DS1922L-F5, Maxim integrated, USA) installed at 9 out of 12 plots at 5 cm depth in the mineral soil at both sites. In the same plots, moisture was measured in the mineral soil at a depth of 5 cm with three EC-5 soil moisture sensors per plot (Decagon Devices Inc., Pullman, WA, USA, validated with gravimetrically measured soil moisture).



### 2.6 Data analysis and statistics

#### 2.6.1 Artificial irrigation and sampling

For annual concentration averages, the four seasons were weighed equally, meaning that for the summer concentration values
from summer 2018 and summer 2019 were averaged. To obtain continuous concentration data for the annual flux estimation,
the exponential relationship with soil temperature was used, expressed as $Q_{10}$ values. The $Q_{10}$ value is a measure of temperature
sensitivity that is exponential and based on biological, chemical, or physicals reaction rates. $Q_{10}$ values were calculated from
the concentration data during the five irrigation events and the soil temperature delta between the samplings using non-linear

regression. Results showed that $Q_{10}$ values averaged to 4.3 and 3.2 for DIP and DIN and 1.8 and 3.2 for DOP and DON,
respectively (SI 3). Out of 360 possible sampled leachates, 12 were missing due to natural disturbances, such as damage to
lysimeters by wild boars or mice. For calculation of annual fluxes, data of the missing 12 samples were extrapolated from the
other two field replicates, considering the ratio between the replicates during the other samplings. For the interpolation of
concentration data, first, the fitting parameters $\beta_0$ and $\beta_1$ were obtained according to Eq. (1), where $T$ (°C) is the soil

temperature at 5 cm depth.

$$y = \beta_0 e^{(\beta_1 T)} \tag{1}$$

Second, $Q_{10}$ values were calculated based on $\beta_1$ according to Eq. (2) (Guelland et al., 2013).

$$Q_{10} = e^{10\beta_1} \tag{2}$$

This was done for each soil depth at each site independently ($Q_{10}$ values and coefficient of determination in SI 3). Nutrient

concentrations were interpolated according to Eq. (3),

$$R_1 = \frac{R_2}{Q_{10}^{\frac{T2-T1}{10}}} \tag{3}$$

where $R_1$ (mg L$^{-1}$) is the predicted daily nutrient concentration in the leachate, $T_2$ (°C) and $R_2$ (mg L$^{-1}$) are temperature and
nutrient concentration measured at the sampling event, $T_1$ (°C) the daily measured soil temperature, and $Q_{10}$ the estimated
temperature dependency. Interpolation was done for each of the five samplings and then averaged. Predicted values for TDN

and TDP differed on average less than 50% from measured concentrations (SI 4).

#### 2.6.2 Estimation of annual fluxes, calculation of balances and stocks

Annual element fluxes (July 2018 to July 2019), being the mass transfer per area unit from one compartment to the next (mg
m$^{-2}$ yr$^{-1}$), were estimated as the sum of daily water volume (L m$^{-2}$) multiplied with their nutrient concentration (mg L$^{-1}$). Daily
water flux was interpolated from daily rainfall measurements, assuming that daily water volume from the zero-tension

lysimeters was directly proportional to daily rainfall quantity (water yields see Table 2). Measured water fluxes were compared
with throughfall records and with water fluxes modeled using BROOK90(R) within the International Co-operative Programme
on Assessment and Monitoring of Air Pollution Effects on Forests (ICP Forests, Table 2). As the plots were watered with an
artificial irrigation solution lacking N and P and, therefore, direct inputs from throughfall were excluded and hence, the fluxes





are rather an underestimate as compared to standard soil solution monitoring. Balances were calculated for the Oe/Oa horizon

and the A horizon as the difference between input (influx from the horizon above) and the output (export flux to the horizon beneath). Stocks were calculated by multiplying soil nutrient concentrations with fine earth densities. Nutrient concentrations were measured on soil samples taken in July 2019. Soil densities in the mineral horizon were determined with a core cutter of 1 L volume and soil densities from the organic layers were taken from Lang et al. (2017).

### 2.6.3 Statistical analysis with linear mixed-effect models

We assessed treatment effects on element concentrations, fluxes, and element ratios using linear mixed-effects models with the *lmer* function from the *lme4* package (Bates et al., 2015). In case of the unbalanced data set of concentrations, *type 3 anova* was used. *p*-values were obtained with the *lmerTest* package (Kuznetsova et al., 2017). Due to non-normal distributed residuals, all tested parameters were log-transformed for the statistical analysis. According to the experimental design, we included site, +N, +P, season, and soil horizon with an interaction between site, +N, and +P as well as an interaction between horizon and

sampling event as fixed effects. Blocks (level of replication, n = 3) and plots (fertilized squares) were used as random effects. For all statistical analyses, fixed effects were considered significant at $p < 0.05$; $p$-values between 0.05 and 0.1 were considered as marginally significant. Error estimates and error bars are standard errors of the means. All analyses were carried out using R version 3.6.3 (R Core Team, 2020).

## 3 Results

### 3.1 Soil temperature and moisture

The two study years, 2018 and 2019, were exceptionally dry and warm. In 2018, annual precipitation amounted to 772 mm at the high-P site and 463 mm at the low-P site, which corresponded to only 68% and 60% of the long-term mean annual precipitation. This was even more pronounced for the period May to October, receiving only 50% and 40% of the long-term average at the high-P and the low-P site, respectively. Accordingly, volumetric water contents reached values below 10% in

both summers (Figure 1). Maximum soil temperatures in summer were 18°C at both sites (Figure 1). In winter, the high-P site experienced longer periods with soil temperatures around 2°C. At the low-P site, winter soil temperatures dropped that low only for few days, and were generally higher than at the high-P site. From July 2018 to July 2019, the average soil temperature was 9.8°C at the low-P site and 7.9°C at the high-P site (Figure 1). There was a statistically significant negative linear relationship between soil temperature and soil moisture at the low-P site ($R^2 = 0.61$, $p < 0.01$) and at the high-P site ($R^2 = 0.28$,

$p < 0.01$).

### 3.2 Patterns of dissolved carbon, nitrogen, and phosphorus in leachates

Total dissolved P concentrations in leachates differed significantly between horizons ($p_{\text{Horizon}} < 0.01$, SI 5), with the highest concentrations in leachates from the Oe/Oa horizons. Despite strongly differing total soil P (Table 1), there was no statistical



difference in TDP concentration between the two sites in none of the horizons ($p_{Site}$ = n.s.). Concentrations of TDP in leachates

varied strongly with seasons ($p_{Season}$ < 0.01, Figure 1), being highest in summer (control means of Oe/Oa horizon: 0.34 and 0.17 mg L$^{-1}$ for the high-P and the low-P site) and lowest in winter (control means of Oe/Oa horizon: 0.03 and 0.04 mg L$^{-1}$ for the high-P and the low-P site). The increased TDP concentrations in summer were mainly due to increased DIP concentrations, exceeding those in winter on average by factor of 20 at the high-P site and by factor of 10 at the low-P site. Concentrations of DOP increased from winter to summer on average only by a factor of 2.5 at both sites. Averaged across sites, DOP constituted

49% of TDP in spring and winter and 26% of TDP in summer and fall. Overall, the contribution of DOP to TDP was higher at the high-P than at the low-P site ($p_{Site}$ = 0.01).

Total dissolved N (TDN) concentrations varied between horizons ($p_{Horizon}$ < 0.01), but not between sites ($p_{Site}$ = n.s. for all horizons). Highest concentrations occurred in leachates from the Oe/Oa horizons. As for TDP, TDN concentrations showed strong seasonal differences ($p_{Season}$ < 0.01, Figure 1), with highest values in summer (control means of Oe/Oa horizon: 5.0 and

3.0 mg L$^{-1}$ for the high-P and the low-P site) and lowest values in winter (control means of Oe/Oa horizon: 0.5 and 1.0 mg L$^{-1}$ for the high-P and the low-P site). This pattern persisted for all N forms ($p_{Season}$ < 0.01), with DIN showing stronger seasonal variations than DON. While DIN concentrations were 9.5- and 3.5-times higher in summer than in winter at the high-P and the low-P site, respectively, DON concentrations increased from winter to summer by a factor of 3.8 at the high-P site and 3.4 at the low-P site.

Concentrations of DOC differed strongly between the two sites ($p_{Site}$ < 0.01, SI 6). They increased from the litter to the A horizon at the high-P site ($p_{Horizon}$ < 0.01). At the low-P site, the highest concentrations occurred in leachates from the Oe/Oa horizon. When averaged, DOC concentrations were twice as high at the low-P site than at the high-P site ($p_{Site}$ < 0.01). The concentrations varied strongly with seasons ($p_{Season}$ < 0.01), and patterns differed between sites and horizons ($p_{Season:Horizon}$ < 0.01). At the low-P site, DOC concentrations in leachates from the Oe/Oa horizons of the control plots were highest in spring

(24.8 mg L$^{-1}$) and lowest in autumn (10.3 mg L$^{-1}$). At the high-P site, the DOC concentrations in leachates from the Oe/Oa horizons of the control plots were highest in autumn (13.3 mg L$^{-1}$) and lowest in winter (1.7 mg L$^{-1}$).





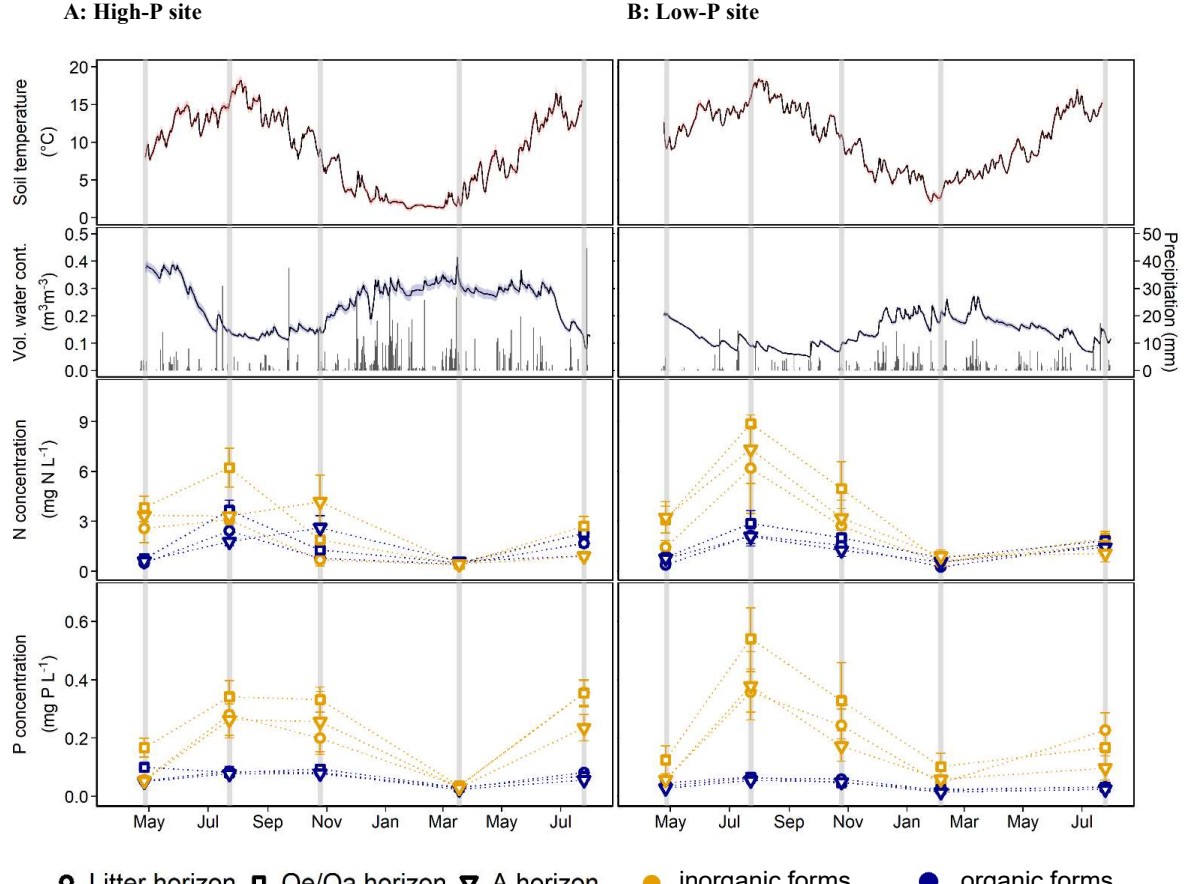

**Figure 1: Soil temperature, soil moisture, and concentration of inorganic and organic nitrogen (N) and phosphorus (P) forms during the experimental duration at the high-P site (A) and the low-P site (B). Row 1 and 2: Soil temperature and volumetric water content**
**measured at 5 cm in the A horizon during the sampling period. Row 3 and 4: Organic and inorganic N and P concentrations in leachates from litter, the Oe/Oa horizon, and A horizon for each site and horizon (including fertilization treatments, n = 12), at five samplings at the two sites. Error bars = standard error. Data for dissolved organic carbon can be found in SI 6.**

### 3.3 Solute concentrations as related to temperature and moisture

The seasonal variations of solute leaching were mirrored by positive relationships of mean DIP, DOP, $NH_4^+$, $NO_3^-$, and DON
concentrations with soil temperature (SI 3), being stronger for P than for the N and for inorganic than for organic forms (higher $Q_{10}$ values for inorganic N and P forms, and higher $Q_{10}$ values for TDP than TDN, SI 3). The temperature−concentration relationships were better reflected by exponential than linear models (higher $R^2$). Mean concentrations were negatively correlated with soil moisture, especially for DON and DIP. Relations were less strong to moisture than to soil temperature.





### 3.4 Annual fluxes of water and solutes

Annual measured vertical water fluxes, averaged over all horizons, amounted 480 L m$^{-2}$ yr$^{-1}$ at the high-P site and 230 L m$^{-2}$ yr$^{-1}$ at the low-P site, corresponding to 57-73% of measured throughfall. They decreased slightly with soil depth (Table 2). Modeling of water fluxes using LWF-Brook90 (done by the Northwest German Forest Research Institute (NW-FVA) and Bayrische Landesanstalt für Wald und Forstwirtschaft (LWF)) gave similar estimates, differing from the measured water fluxes by 4 to 79 L m$^{-2}$ yr$^{-1}$ (+2 to +19% of measured flux) at different depths at the low-P and the high-P site, respectively (Table 2).


**Table 2: Annual water fluxes measured by zero-tension lysimeters below the litter horizon, the Oe/Oa horizon, and the A horizon at the high-P site and the low-P site, from July 2018 to July 2019. For comparison, the modeled fluxes from the same sites (both ICP Forest Level II sites) are presented. Details in SI 7.**

| Site (Time period: 24.07.2018-24.07.2019) | Specification | Annual water flux [L m$^{-2}$ yr$^{-1}$] |
|---|---|---|
| High-P site - ICP Forest site[+] | measured throughfall | 722 |
| High-P site** | Litter | 525 |
| High-P site** | Oe/Oa horizon | 507 |
| High-P site** | A horizon | 411 |
| High-P site - ICP forest site*[+] | Vertical matrix flow modeled av. 5-6 cm | 583 |
| High-P site - ICP forest site*[+] | Vertical matrix flow modeled av. 1-5 cm | 553 |
| High-P site - ICP forest site*[+] | Vertical matrix flow modeled -5 cm | 489 |
| Low-P site - ICP forest site[+] | measured throughfall | 364 |
| Low-P site ** | Litter | 244 |
| Low-P site ** | Oe/Oa horizon | 239 |
| Low-P site ** | A horizon | 210 |
| Low-P site - ICP forest site*[+] | Vertical water flow modeled at 0 m | 249*** |
| Low-P site - ICP forest site*[+] | Vertical water flow modeled at 0.2 m | 233*** |

\* modeled with LWF-Brook90(R) model

\*\* includes average volume of art. irrigation

\*\*\* time period: modeled only until 14.07.2019

[+] Data source: Personal comm. Northwest German Forest Research Institute (NW-FVA) and Bayrische Landesanstalt für Wald und Forstwirtschaft (LWF)

**Figure 2: Estimated annual fluxes of dissolved inorganic and organic nitrogen (DIN and DON) and phosphorus (DIP and DOP) from in the litter horizon, the Oe/Oa horizon, and the A horizon at the high-P site and the low-P site, as affected by N addition (+N), P addition (+P), and the combined N and P addition (+N+P), as compared to the control (Cont.). The annual flux was estimated for the time period July 2018 to July 2019. Data for dissolved organic carbon can be found in SI 6.**



The annual TDP fluxes in the control plots from all horizons ranged between 12 and 60 mg m$^{-2}$ yr$^{-1}$. They were similar at both

sites ($p_{Site}$ = n.s. for all horizons), with lowest values in the A horizons ($p_{Horizon}$ < 0.01). The A horizons at both sited were P sinks, as the balances (the difference between the fluxes into and out of a given horizons) were positive (Table 3). At the high-P site, the Oe/Oa horizon was a source for P, exhibiting a negative balance, while at the low-P site the Oe/Oa horizon was a P sink (Table 3). The TDP fluxes from the litter layer corresponded to 17% of the litter P stock at the high-P site and 9% of litter P stock at the low-P site (Table 3). The portion of P stock leached decreased with soil depth. Fluxes of TDP from the A horizon,

represented 0.1% of the horizon's P stock at the high-P site and 0.2% at the low-P site (Table 3). Annual DIP fluxes in the control plots did not differ between sites, whereas DOP fluxes were higher at the high-P than at the low-P site ($p_{Site}$ < 0.01). From the Oe/Oa horizon, mean DOP fluxes in the control plots were 20 mg m$^{-2}$ yr$^{-1}$ at the high-P site and 6 mg m$^{-2}$ yr$^{-1}$ at the low-P site. The contribution of DOP to DTP was more similar across sites, being on average 33% at the high-P site and 21% at the low-P site.

**Table 3: Comparison of phosphorus (P) fluxes from litter, Oe/Oa, and A horizons with respective stocks (average of three field replicates taken in July 2019) of each horizon at the high-P site and the low-P site. Balances: the difference between the fluxes into and out of a given horizon; positive values reflect net accumulations, negative values net losses. Data from unfertilized plots (n = 3). Means ± standard error (SE). Data for dissolved organic carbon can be found in SI 6.**

| Site | Horizon | P stock of control | TDP flux of control | DIP flux of control | DOP flux of control | Annual TDP flux as % of stock | TDP balance |
|------|---------|---------|---------|---------|---------|---------|---------|
|      |         | mg P m$^{-2}$ | mg m$^{-2}$ yr$^{-1}$ | mg m$^{-2}$ yr$^{-1}$ | mg m$^{-2}$ yr$^{-1}$ | % yr$^{-1}$ | mg P m$^{-2}$ |
| **High-P** | Litter | 290 | 49.7 (±4.6) | 26.6 (±3.1) | 22.1 (±2.4) | 17 | |
| **High-P** | Oe/Oa | 4600 | 59.9 (±8.1) | 39.5 (±6.2) | 19.9 (±2.3) | 1.3 | -10.2 |
| **High-P** | A | 60500 | 39.2 (±14.5) | 25.5 (±12.1) | 13.6 (±3) | 0.1 | +20.7 |
| **Low-P** | Litter | 630 | 43.4 (±2.1) | 32.9 (±1.5) | 10.5 (±1.5) | 6.9 | |
| **Low-P** | Oe/Oa | 6350 | 28.2 (±13.2) | 22.3 (±12) | 5.8 (±1.3) | 0.4 | +15.2 |
| **Low-P** | A | 6340 | 12.2 (±3.8) | 8.7 (±3.3) | 3.6 (±0.6) | 0.2 | +16.0 |

**Table 4: Comparison of nitrogen (N) fluxes from litter, Oe/Oa, and A horizons with respective stocks (average of three field replicates taken in July 2019) of each horizon at the high-P site and the low-P site. Balances: the difference between the fluxes into and out of a given horizon; positive values reflect net accumulations, negative values net losses. Data from unfertilized plots (n = 3). Means ± standard error (SE). Data for dissolved organic carbon can be found in SI 6.**

| Site | Horizon | N stock of control | TDN flux of control | DIN flux of control | DON flux of control | Annual TDN flux as % of stock | TDN balance |
|------|---------|---------|---------|---------|---------|---------|---------|
|      |         | mg N m$^{-2}$ | mg m$^{-2}$ yr$^{-1}$ | mg m$^{-2}$ yr$^{-1}$ | mg m$^{-2}$ yr$^{-1}$ | % yr$^{-1}$ | mg N m$^{-2}$ |
| **High-P** | Litter | 5110 | 449.1 (±41) | 249.7 (±35.3) | 201.0 (±7) | 8.8 | |
| **High-P** | Oe/Oa | 38900 | 733.8 (±81.8) | 478.7 (±46.5) | 257.4 (±36.8) | 1.9 | -285 |
| **High-P** | A | 240000 | 624.9 (±101.8) | 462.3 (±78.6) | 211.6 (±27.8) | 0.3 | +109 |
| **Low-P** | Litter | 11900 | 716.9 (±129.2) | 458.5 (±137.1) | 257.5 (±25.4) | 6.0 | |
| **Low-P** | Oe/Oa | 141000 | 652.9 (±340.4) | 408.7 (±258.4) | 238.3 (±84.6) | 0.5 | +63.9 |
| **Low-P** | A | 128000 | 292.4 (±104.2) | 199.6 (±85.3) | 93.7 (±17.8) | 0.2 | +361 |



Effects of fertilization on fluxes of TDP and DIP differed among sites, with a comparable pattern for all horizons ($p_{Site:N:P}$ = 0.05 and 0.07, Figure 2). At the low-P site, TDP leaching from the Oe/Oa horizon was increased by separate N and P additions ($p_P$ = 0.07; Oe/Oa horizon: +198%, and 156% compared to leaching in the control plots), but not when combined. At the high-P site, +N, +P, and +N+P fertilization increased total P leaching (Oe/Oa horizon: +33%, +51%, and +75% compared to the control). However, only the increase for the N×P treatment was statistically significant ($p_{Site:N:P}$ = 0.07). Fertilization did not

affect DOP, at neither site nor soil depth ($p_{Site:N:P}$ = n.s., Figure 2).

Fluxes of TDN in control plots ranged between 292 and 734 mg m$^{-2}$ yr$^{-1}$, and were similar at the two sites ($p_{Site}$ = n.s. for all horizons) and differed between horizons ($p_{Horizon}$ = 0.01). The high-P site had the smallest fluxes in the litter layer, while at the low-P site the smallest fluxes occurred in the A horizon (Table 4). Since N-free rainwater was applied, the fluxes basically reflect N release from soils. The average contribution of DON to TDN was 36% at both sites. There were differences between

sites in distribution of NO$_3^-$ and NH$_4^+$. At the high-P site, 20% of TDN leached was NH$_4^+$ and 44% NO$_3^-$. The contributions of NH$_4^+$ and NO$_3^-$ were each 30% at the low-P site. Fertilization affected leaching at sites and horizons in a similar way: N addition increased the TDN leaching ($p_N$ < 0.01); P addition alone did not affect leaching ($p_P$ = 0.40) (Figure 2); N×P addition increased leaching at the high-P, but not at the low-P site.

Annual DOC fluxes in control plots ranged between 1.68 and 3.40 g m$^{-2}$ yr$^{-1}$ (SI 6). They were higher at LUE, and largest in

the Oe/Oa horizon ($p_{Site}$ = 0.02, $p_{Horizon}$ < 0.01) at both sites. Fertilization affected DOC fluxes at the sites differently ($p_{Site:N}$ = 0.05). At the high-P site, DOC leaching increased from all horizons upon N fertilization. At the low-P site, fertilization only affected the A horizon, where DOC leaching was increased by all fertilization treatments ($p_{N:P}$ = 0.11) on average by factor 1.6 compared to the control treatment.

**3.5 Stoichiometry of inorganic and organic nitrogen and phosphorus leaching**

Ratios of TDN:TDP and DIN:DIP in leachates and soil solutions increased with soil depth from litter to A horizon ($p_{Horizon}$ < 0.1 and $p_{Horizon}$ = 0.1). On average, ratios were greater at the low-P than at the high-P site ($p_{Site}$ = 0.09 and 0.07). Concentrations of DOC and DOP correlated significantly, with stronger correlations at the high-P site than at the low-P site. Ratios of DOC:DOP in control plots ranged between 85 and 605 (SI 8). Mean DOC:DON ratios ranged from 7 to 21, and mean DON:DOP ratios from 9 to 37 (SI 8). The ratios were significantly higher at the low-P than at the high-P site ($p_{Site}$ < 0.01),

paralleling the C:N and C:P ratios in soil (Table 1). However, the correlations between ratios in leachates and soils were not significant (SI 9). Stocks of $P_{Resin}$, as a measure of available P, did only correlate with leached DIP in autumn (October 2018, $R^2$ = 28, $p$ = 0.06) and winter (Feb./Mar 2019, $R^2$ = 33, $p$ = 0.03). All element ratios in leached DOM showed a pronounced seasonality ($p_{Season}$ < 0.01) with smaller DOC:DON ratios and DOC:DOP ratios, but higher DON:DOP ratios in summer and fall (Figure 4). DIN:DIP ratios were highest in spring at both sites (Figure 4).








**Figure 3: Temporal variations in ratios among dissolved organic carbon (DOC), nitrogen (DON), and phosphorus (DOP) as well as dissolved inorganic N (DIN) and P (DIP) in leachates from the Oe/Oa horizon at the high-P site and the low-P site. The same seasonal patterns were observed for the litter and the A horizon. Data averaged over fertilization treatments, boxplots depict variation from three field replicates. Wint.19 = February/March 2019.**





Nitrogen addition decreased DOC:DON ratios significantly ($p_N < 0.01$, Figure 5), and increased the DON:DOP and DIN:DIP ratios ($p_N < 0.01$). Fertilization with P had no effect at the high-P site, but caused decreases in DON:DOP and DIN:DIP ratios at the low-P site ($p_{Site:P} = 0.04$ and $p_{Site:N:P} = 0.06$).


**Figure 4: Ratios of dissolved organic carbon (DOC), dissolved organic and inorganic nitrogen (DON and DIN), and dissolved organic and inorganic phosphorus (DOP and DIP) in leachates from the Oe/Oa horizon as affected by +N, +P, and +N+P fertilization at the**



**high-P and the low-P site. Data averaged before over seasons (seasons were equally weighted) and the boxplots depict variation from the three field replicates.**

**4 Discussion**

**4.1 Seasonal patterns**

In agreement with our hypothesis, DIP concentrations varied more strongly with seasons than DOP. Concentrations of DIP represent the net result of release and retention processes. Seasonal variations can be attributed to changes (i) in mineralization rates of SOM, (ii) in demand and uptake by plants and microorganisms, (iii) in sorption kinetics, or (vi) due to the rewetting

by the irrigation, especially on dry soil during summer. Demand and uptake of phosphate are greatest in summer, and consequently the activity of phosphomonoesterases were found to be substantially higher in summer than in winter in the same soils (Fetzer et al., 2021). Also, desorption of phosphate increases with temperature and sudden rewetting of dry soil can cause desorption as well (Barrow, 1983), both potentially contributing to higher DIP concentrations in summer. However, we assume that P from lysed microbial cells following drying-rewetting strongly contributed to the summer peak of P concentrations. In

support, there was a strong decline of DOC:DON ratios and DOC:DOP ratios from spring to summer 2018, and a smaller summer peak of DIN than of DIP in forest floor leachates. The decline in nutrient ratios mirror a higher share of microbial metabolites in these leachates since released microbial cytoplasma has very narrow C:N:P ratios of 12:3:1 at the same sites (Siegenthaler et al. (2021, submitted)). The lysis of microbe al cells leads to a pulse of N and P from microbial metabolites (Gao et al., 2020; Schimel, 2018). In our study, the N and P summer peak was more pronounced at the low-P site, which

experienced a longer and stronger dry phase than the high-P site and resulted in a drop of DOC:DON ratios from 28 in spring to ratios below 10 in summer. Artificial irrigation during the dry summer 2018 with 20 mm h$^{-1}$ could have promoted the drying and rewetting effect by rapidly leaching DIP and DOP, reducing the time for a biological uptake or sorption. This effect was not observed in a microcosm study, where soils of the same sites were subjected to drying and rewetting (Gerhard et al., 2021). However, in their study, the drying was only moderate and the focus was laid on the mineral soil with lower SOM contents,

and hence, less microbial biomass that can be released upon drying and rewetting.

In contrast to DIP, DOP shows similar concentrations throughout the year, and hence a low temperature dependency. In agreement, DON also varied less with season than DIN. We relate the low temperature effects to the cancelling out of the production and mineralization of DOP and DON, which - as they are both microbial processes - are temperature sensitive. An analogous conclusion has been drawn in soil warming studies in the laboratory and field, where leaching of DOC – correlating

with DOP in our study – showed smaller responses to temperature than respiratory processes (Gödde et al., 1996; Hagedorn et al., 2010; Müller et al., 2009).





### 4.2 Link of leaching rates to site properties

We have expected that TDP fluxes would be higher in the high-P than in the low-P soil. Contrasting our hypothesis, there was no statistically significant difference in TDP fluxes between the sites in all horizons. The surprisingly negligible site effects could result from sorption that could have balanced out differences in the P release. Generally, phosphate can sorb to charged surfaces which represents a dominant retention mechanism in mineral soils (Barrow, 1983; Berg and Joern, 2006; Rechberger et al., 2021). Sorption seemed particularly strong in the mineral soil at the high-P site, which had higher contents of clay and aluminum and iron oxides that provide more sorptive mineral surfaces than the sandy low-P site with bleached quartz grains (Lang et al., 2017). The stronger sorption at the high-P site is supported by the smaller proportion of the annual TDP flux compared to the P stock in the A horizon in the high-P site (Table 3). Sorption might also occur in the organic horizons, and especially in the Oa horizons, as they can contain minerals, due to aeolian deposition or bioturbation, with the latter being more pronounced at the high-P site showing signs of a high faunal activity. Phosphate can also sorb via cation bridges in the organic horizons (Gaume et al., 2000; Gerke and Hermann, 1992; Rechberger et al., 2021). Phosphate competes in organic horizons to substantially smaller amounts of sorptive minerals, largely occupied by organic matter (Rechberger et al., 2021). In result, sorption in the organic horizons is less important than in mineral horizons. Nevertheless, our assessment showed that the thick Oe/Oa horizon was a sink for P at the low-P site (Table 3), which we primarily relate to biotic uptake. However, unexpectedly, also in the litter and the Oe/Oa horizon, P leaching did not statistically differ between sites. This finding can also be explained by a cancelling out of the potentially higher P release from the organic layer at the high-P site (reflected in higher $P_{Resin}$ contents) by the greater thicknesses of the organic layers at the low-P site and hence a greater reservoir from which P can be mobilized. We presume that the greater P stock in the organic layers at the low-P soil (Table 3) is an inherent site property linked to low-P sandy parent material that promoted accumulation of organic material on top of mineral soils due to low biological activity (Hauenstein et al., 2018). Consequently, P fluxes in relative terms - when compared to P stocks - were larger from both organic horizons in the high-P soil than in the low-P soil (Table 3) due to thinner horizons. In contrast to the organic layers, P leaching from A horizons relative to P stocks was twice as high in the low-P than in the high-P soil (Table 3). The smaller P leaching from the A horizon at the high-P site will foster the difference in P storage between the two soils in the long-run.

In contrast to TDP fluxes, we expected higher contribution of DOP to total P leaching at the low-P site, because DOP, as part of DOM, is leached during SOM processing, while phosphate is assumed to be more efficiently retained by plant and microbial uptake in low-P forest ecosystems (Hedin et al., 2003). However, opposed to these expectations, the absolute DOP fluxes as well as their contribution to total P fluxes were higher in the high-P soil than in the low-P soil. We relate this finding to the higher soil organic P content at the high-P site (Table 1), resulting in smaller C:P ratio of SOM in all horizons, which translate also into smaller DOC:DOP ratios.





### 4.3 Fluxes

Despite of using an irrigation approach to overcome site and weather variations, the estimated TDP fluxes, ranging between
12 and 60 mg total P m$^{-2}$ yr$^{-1}$ across all horizons (Table 3), compare well with the P fluxes measured in other forest ecosystems, ranging from 9 to 62 mg P m$^{-2}$ yr$^{-1}$ (Qualls, 2000; Fitzhugh et al., 2001; Hedin et al., 2003; Piirainen et al., 2007; Sohrt et al., 2019; Rinderer et al., 2021). The P fluxes observed here are about one magnitude lower than those determined in a laboratory study with isolated horizons from the same sites (ranging from 70 to 320 mg P m$^{-2}$ yr$^{-1}$ across all horizons; Brödlin et al., 2019a). The higher P release in the laboratory than in situ can be attributed to the regular leaching, the longer as well as more
complete contact of soil with excessive artificial rainwater, and the lack of uptake by plants, rather reflecting potential release rates. The comparison between these two approaches demonstrates that in soil continuous release and immobilization of P takes place, but only a small proportion of released P becomes eventually leached.

**How relevant are these P leaching fluxes?** The P export from the Oe/Oa horizon observed in the present study is in the same order of magnitude of reported P inputs with bulk precipitation or throughfall at another German beech forest (60 mg P m$^{-2}$ yr$^{-}$
$^1$, Sohrt et al., 2019). In Germany, atmospheric P inputs may largely originate from fertilized agricultural land, but information on P deposition in general still is scarce (Bol et al., 2016; Tipping et al., 2014; Vogel et al., 2021; Wang et al., 2014) and not available for the two study sites. The P fluxes from the A horizon at the high-P and at the low-P site are approx. 154% and 47% of reported atmospheric P deposition in Germany. In comparison, N fluxes from the A horizons are 30% and 28% of atmospheric N depositions that were measured at the same high-P and the low-P site (Brumme et al., 2021; NW-FVA, 2020).
This lends support that leaching reinforces the nutritional imbalance between N and P due to N enrichment induced by the divergent atmospheric inputs of the two elements (Peñuelas et al., 2013).

Since the P fluxes decreased towards the A horizon, they likely decrease further in deeper mineral soils due to lower release of P and higher retention of DOP and DIP by sorption to reactive minerals (Barrow, 1983; Berg and Joern, 2006; Brödlin et al., 2019a) as well as uptake by roots and microorganisms in subsoils. Consequently, P leaching losses from entire soil profiles
are likely even smaller than from the topsoil observed here. Therefore, P losses form the entire soil profile likely do not exceed P depositions, contradicting the idea of these forest ecosystems to deplete in P due to leaching, as suggested by decreasing leaf P concentrations in beech trees (Lang et al., 2017; Talkner et al., 2015). The estimated P fluxes from the Oe/Oa horizon are even one order of magnitude lower than the P input via annual litterfall, which amounts to 229 mg P m$^{-2}$ yr$^{-1}$ at the high-P site and 156 mg P m$^{-2}$ yr$^{-1}$ at the low-P site (Lang et al., 2017). This implies that a large proportion of litterfall P must be taken up
by plants and microorganisms. However, without the current "P pollution", and at the centennial and millennial time scale, relevant for soil and ecosystem development, the small P leaching in the A horizon, with 0.1 and 0.2% of its P stock being leached during one year (Table 3), is important for redistributing P within the soil profile. When leached from the entire profile, or exported via lateral flow (Rinderer et al., 2020), it may also contribute to the depletion of the ecosystems in P (e.g. Hedin et al., 2003; Richardson et al., 2004)



### 450   4.4 Fertilization effects

**Phosphorus fluxes**. In agreement with our hypothesis, fertilization had stronger effects on the leaching of inorganic than on organic P and N. Also, the fertilization effects were stronger at the low-P than at the high-P site. However, counterintuitively, DIP leaching increased more strongly upon fertilization with N than with P at the low-P site. We relate this to the overcoming of N limitation for the production of enzymes hydrolyzing organic P. Nitrogen-limited phosphatase production is also indicated

by the increased activity of phosphomonoesterases after N addition in the studied leachates and soil solutions (Fetzer et al., 2021). Similar effects have been reported for grassland soils (Widdig et al., 2019) and in mesocosm experiments with soils from the low-P site (Holzmann et al., 2016). Also, Siegenthaler et al. (2021, submitted) showed that N addition was the main factor driving changes in bacteria and fungi communities at the low-P site. Finally, N fertilization might have stimulated overall microbial processing of SOM (Griepentrog et al., 2014; Hagedorn et al., 2012), which is supported by observed increased

DOC leaching at the high-P site (SI 6).

Although the amount of added P was much higher than that of N when compared to soil N and P stocks (Table 5 and 6), P fertilization affected P leaching less than N fertilization. In the Oe/Oa horizon, P addition was 109 and 79% of the P stock at the high-P and the low-P site, while N addition was 39 and 11% of the N stock at the high-P and the low-P site. Nevertheless, N leaching increased more than P leaching (Table 5 and 6). Most probably, a large fraction of added P was either rapidly

sorbed to reactive minerals, taken up by plants (Hauenstein et al., 2020), or immobilized by soil microorganisms. Microbial biomass in the Oe/Oa horizons at two study sites had very low microbial C:P ratios of 12 before the nutrient addition (Siegenthaler et al. (2021, submitted). Fertilization with P did not affect the microbial P in the Oe/Oa horizons at the high-P site, but increased it at the low-P site with the +N+P treatment compared to the +N treatment (Siegenthaler et al., 2021, submitted), strongly suggesting microbial immobilization of added P. In temperate German beech forests of the same region,

Zederer et al. (2017) found that in Oe/Oa horizons with average soil C:P ratios of 390, microbial biomass had C:P ratios of 13 and comprised approx. 30-50% of total P. This indicates that organic layers have a high capacity to retain microbially-bound P. Our leaching study shows that very little of retained P is re-released into soil solution, indicating that P retention is effective in the organic layer and not only in mineral soils, where sorption is higher.

In the longer term, however, continuous P retention in the organic layer, presumably by microbial P immobilization, decreases

soil total C:P ratios as already observed for the Oe/Oa horizon at the low-P site. Here, the three year-long P fertilization decreased the measured soil total C:P ratios from 597 (Control) to 354 (P addition, SI 8) and P stocks increased by 0.2 and 1.8 g P m$^{-2}$ in the litter and in the Oe/Oa horizon (Table 5). The decline in soil C:P ratios might enhance net P release and leaching, as observed in leaching studies with the same soils by Brödlin et al. (2019a). Although the low-P soil is more retentive for P due to its higher soil total C:P ratios and hence, a higher P demand by plants and microorganisms, effects of P fertilization on

P leaching were greater at the low-P than at the high-P site (+156 vs. +51% in the Oe/Oa horizon, Table 5). We attribute this apparent conflict to the low N availability at the low-P site, and thus, a possible co-limitation by N, which reduced the biological P uptake.



The effects of the combined addition of N and P on P leaching depended on the site. It only increased P leaching at the high-P site. Here, the effects of fertilization with N and P were largely additive but not synergistic, i.e. the positive effect of N addition (+33% in the Oe/Oa horizon), presumably by supporting increased phosphatase activity, and the enhanced P leaching with P fertilization (+51% in the Oe/Oa horizons), resulted in an overall increase of 75%. In contrast, at the low-P site, there was an antagonistic interaction as the combined addition did not affect P leaching as compared to the control and even reduced it as compared to the fertilization with N only. This is in accordance with results by Siegenthaler et al. (2021), who observed increased microbial P at the low-P site, but only for the +N+P treatment. Here, we can only speculate about the mechanisms behind this finding as P leaching is the net product of various counteracting mechanisms. One reason could be that only the combined addition removed nutrient limitation and promoted an increase in microbial biomass at the low-P site, which might have increased microbial P immobilization, and thus, reduced P leaching. Laboratory fertilization experiments with mineral soils from the same sites indicated mostly microbial C and P limitation for the low-P soil, and N and N-P co-limitation for the high-P soil (Chen et al., 2019; Rodionov et al., 2020). However, a study on P concentrations in xylem sap on the fertilized plots likewise observed the antagonistic interaction on P concentrations in xylem after the +N+P treatment at the low-P site, but a significant synergistic effect of the combined +N+P fertilization at the high-P site (Hauenstein et al., 2020), supporting the theory of a co-limitation of N and P at the low-P site.

**Table 5: Soil phosphorus (P) stocks from the control plots (n=3) at the high-P and the low-P site for the litter and the Oe/Oa horizon as well as added amounts P, and the change in annual leaching of total dissolved P (TDP) and in element stocks due to fertilization. Other treatments given in SI 10.**

| Site: horizon | P stock from control plot | P fertilization for whole profile | Fertilization to stock | Change in the P stock due to P fertilization | Change of TDP leaching due to P fertilization |
|---|---|---|---|---|---|
| | g P m⁻² | g P m⁻² | % | g P m⁻² | mg P m⁻² yr⁻¹ (% to control) |
| **High-P: Litter** | 0.3 | | 1728 | 0.1 | 4.3 (9%) |
| **High-P: Oe/Oa** | 4.6 | | 109 | -1.5 | 30.5 (51%) |
| **High-P: down to 1 m soil depth*** | 904 | 5 | 0.6 | | |
| **Low-P: Litter** | 0.6 | | 791 | 0.2 | 13.3 (31%) |
| **Low-P: Oe/Oa** | 6.3 | | 79 | 1.8 | 43.9 (156%) |
| **Low-P: down to 1 m soil depth*** | 164 | 5 | 3.0 | | |

* From Lang et al., 2017, soil and forest floor



**Table 6: Soil nitrogen (N) stocks from the control plots (n=3) at the high-P and the low-P site for the litter and the Oe/Oa horizon as well as added amounts N, and the change in annual leaching of total dissolved N (TDN) and in element stocks due to fertilization. Other treatments given in SI 10.**

| Site: horizon | N stock from control plot | N fertilization for whole profile | Fertilization to stock | Change in the N stock due to N fertilization | Change of TDN leaching due to N fertilization |
|---|---|---|---|---|---|
| | g N m$^{-2}$ | g N m$^{-2}$ | % | g N m$^{-2}$ | g N m$^{-2}$ yr$^{-1}$ (% to control) |
| **High-P: Litter** | 5.11 | | 294 | 0.1 | 0.58 (129%) |
| **High-P: Oe/Oa** | 38.9 | | 39 | 0.4 | 0.86 (118%) |
| **Low-P: down to 1 m soil depth*** | 1300 | 15 | 1.2 | | |
| **Low-P: Litter** | 11.9 | | 126 | 1.4 | 0.38 (53%) |
| **Low-P: Oe/Oa** | 141 | | 11 | 35.2 | 0.74 (114%) |
| **Low-P: down to 1 m soil depth*** | 700 | 15 | 2.1 | | |

* From Lang et al., 2017, soil and forest floor

**Nitrogen fluxes**. In contrast to DIP, DIN leaching responded consistently positive to N fertilization (Table 6), which signifies that the N additions led to an N surplus in both soils. Phosphorus addition did not affect N leaching, indicating that the release of N is not related to P availability. This is in accordance with N fertilization experiments in grasslands (Schleuss et al., 2021). Although to a lower extent than for DIN, N fertilization increased DON concentrations, which went along with significant decreases in DOC:DON ratios (Figure 4). This decline is consistent with findings of lower C:N ratios in extracted OM after +N+P fertilization in grasslands (Neff et al., 2000), suggesting that added N was immobilized in the solubilizable SOM pool (Schleuss et al., 2021), and/or that N fertilization might have increased the contribution of microbial metabolites with low C:N ratios to DOM. In support, N fertilization of a young beech and spruce forest promoted the production of new fungal residues in the soil and hence fungal turnover (Griepentrog et al., 2014), potentially leading to the release of fungal metabolites.

## 5 Conclusions

Firstly, it turned out that climatic conditions were the strongest drivers of N and P leaching. In the exceptionally hot and dry summer 2018, DIP concentrations were particularly high, while DOC:DON and DOC:DOP ratios were extraordinary low, which strongly suggests N and P release from lysed cells upon drying-rewetting. This effect was stronger at the low-P site. Dissolved organic and inorganic compounds were differently affected by seasons, especially in the case of P, with DIP showing seasonal patterns, but not DOP. These results suggest that changes in climate towards more frequent extremes can alter stoichiometries in leachates of forest soils, causing accelerated DIP leaching.



Secondly, the differences in P leaching between sites were smaller than expected. The organic layers at the low-P soil with
lower P concentrations had similar P fluxes than the high-P soil, as the pool sizes of the organic layer were larger under low-
P conditions and "compensated" for the lower P release per unit organic matter. In the mineral soil, the high-P site had more
charged surfaces and hence, stronger sorption capacity that likely influenced the magnitude of leaching. Phosphorus leaching
from the A horizons relative to P stocks was twice as high in the low-P than in the high-P soil, fostering the difference in P
storage between the two soils in the long-run.

Thirdly, our estimated P fluxes were comparable to reported atmospheric P inputs, implying that these forest ecosystems likely
to not deplete in P by leaching as long as there are atmospheric P inputs in similar orders of magnitudes. Phosphorus fluxes
from the A horizons, relative to reported P deposition, were much higher than the N fluxes, relative to reported N depositions,
which lends support that leaching contributes to the nutritional imbalance between N and P due to divergent atmospheric inputs
of the two elements.

Fourthly, we showed that intrinsic soil N and P availability determined the effect of N and P addition on nutrient release and
leaching. Fertilization effects were additive at the high-P site, with significant increases in P leaching only after +N+P addition.
At the low-P site, fertilization effects were antagonistic, with only +N and +P leading to an increase in P leaching, hinting at
a N and P co-limitation of microorganisms at the low-P site. Additionally, effect sizes of the increase in leaching due to nutrient
addition were higher in the low-P system, and hence, the low-P ecosystem was more responsive to fertilization than the high-
P system. Overall, this implies that low-P ecosystem is likely more vulnerable to environmental future changes.





**Data availability**

All datasets for this study are online at EnviDat: https://doi.org/10.16904/envidat.234

**Author contributions**

FH, KK, EF, JF contributed to conception and design of the study. JF and FH conducted the field experiments. JF and KK did the analytical measurements. JF did the data analyses, the data visualization, and wrote the first draft of the manuscript. All authors contributed to the interpretation of the findings and to the manuscript revision, they read and approved the submitted version.

**Competing interests**

The authors have the following competing interests: Frank Hagedorn is associate editor of BG.

**Acknowledgements**

We gratefully acknowledge the financial support by the Swiss National Science Foundation (SNF) (project number 171171) that supports JF and FH, as well as the German Research Foundation (DFG) that funded the Priority Program SPP 1685
''Ecosystem nutrition: Forest strategies for limited phosphorus resources'' (grant KA1673/9-2) supporting JF and KK. We thank all people involved in establishing and maintaining the monitoring sites ("Bayerische Landesanstalt für Wald und Forstwirtschaft" (LWF) and "Nordwestdeutsche Forstliche Versuchsanstalt" (NW-FVA)) as well as the fertilization experiment (J. Krüger and many more). Many thanks to R. Köchli for great help during the many field trips and D. Brödlin, A. Missong, C. Schmidt-Cotta, I. Vögtli, L. Jansing, and D. Kaiser for support in the field. We thank the WSL central laboratory
(A. Schlumpf, K. v. Känel, J. Bollenbach, U. Graf, D. Pezzotta) and the WSL forest soil laboratory (A. Zürcher, B. Rahimi, D. Christen) for chemical analyses and support. A. Boritzki and P. Winkler at the Halle soil laboratory and L. Schönholzer at the Eschikon laboratory are gratefully acknowledged for phosphorus and carbon analyses. We thank Maja Siegenthaler for the $P_{Resin}$ data.





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
