# Peer review of "Leaching of inorganic and organic phosphorus and nitrogen in contrasting beech forest soils – seasonal patterns and effects of fertilization"

_Biogeosciences, 2021_

## Author Comment (AC1)

Reply to the Reviewers

**Response to Reviews of: Leaching of inorganic and organic phosphorus and nitrogen in contrasting beech forest soils – seasonal patterns and effects of fertilization**

**Review #1**

The Ms of Fetzer et al. entitled 'Leaching of inorganic and organic phosphorus and nitrogen in contrasting beech forest soils – seasonal patterns and effects of fertilization' quantifies annual organic and inorganic P and N fluxes from organic layers and from the mineral topsoil. For this purpose, zero-tension-lysimeters were used in the three soil horizons that were artificially irrigated to standardize water flow. The authors established a comparative study; two sites under beech with different phosphorous availability and sorption capacity (sandy soil and a soil on basalt) were selcted, and both sites were subjected to a full factorial N×P fertilization experiment. During the 18-months monitoring period, the sites were samples five times.

In the context of increasing nutrient imbalances in trees and the occurrence of more frequent and intense climate extremes, the topic of this manuscript is of great importance to both science and practice. The experimental design is state of the art. I like very much that the authors have established a comparative study. The novelty of the study is that the dissolved inorganic and organic N and P fluxes are compared at different nutrient availability of the soils. I think this is not often done. The manuscript is well structured and very well written. After the presentation of the results, the hypotheses are discussed in detail on the basis of the results found and with reference to other studies. The conclusions are clear and based on the results of this study. I recommend the publication of this study in Biogeosciences with minor revisions.

Thank you very much for this positive feedback.

**Specific comments**

95-100          Please provide information to the humus type and to the stand characteristics

**Comment author**: We added the missing information in lines 93, 95, and 100 as following:

"The study was conducted in two mature beech forest stands in Germany with contrasting parent material and P availability. **The stands are dominated by 120-140 year old *Fagus sylvatica* (Lang et al., 2017).** The soil with high P stock (Table 4) is a loamy Cambisol developed on basalt at Bad Brückenau (BBR, 809 m a.s.l., 50.35° N, 9.27° E, referred to "high-P site") **and a mull-like moder organic forest floor layer (Lang et al., 2017).** The soil with the lower P stock is a sandy Cambisol, featuring a thicker organic layer and initial podzolization at Unterlüss (LUE, 115 m a.s.l., 52.8° N, 10.3° E, referred to "low-P site") that developed from glacial till. **The organic forest floor layer is a mor-like moder (Lang et al., 2017)."**

120          Zero-tension-lysimeter: Is the setup similar to Makowski et al., 2020 JPNSS? If yes, you may refer to it because this paper provides more detailed information.

**Comment author**: Correct, the lysimeters we used in the mineral soil (lowest depth) are similar to those used by Makowski et al.. We added the reference to this paper in line 131: "…, having a design similar to those used and described in greater detail by Makowski et al., 2020b."

435         You argue that P leaching reinforces nutrient imbalance between N and P. I am not sure if that can be concluded based on your study. Please explain in more detail your conclusion regarding the contribution of P leaching to nutrient imbalances.

**Comment author**: We relate the N and P leaching to the atmospheric inputs of the two elements, which shows that P leaching as compared to P deposition is higher than for N. To clarify our argumentation, we rephrased the sentence in the lines 483f as follows:

"Therefore, it seems that the leaching losses relative to atmospheric inputs are greater for P than for N, which likely fosters the nutrient imbalances between N and P (Peñuelas et al., 2013)".

440         Julich et al. (2017 in Forests) quantifies the P export from a small forested headwater-catchment in the Eastern Ore Mountains; their annual fluxes were between 2 and 4 mg m-2 and year. These data support your argument.

**Comment author**: Thank you for pointing out that reference, it indeed supports well our discussion. We added it in line 487ff as follows:

"This is supported by annual export rates of P measured in the runoff of forested catchments in Germany of 2-9 mg P $m^{-2}$ $yr^{-1}$ (Julich et al., 2017; Sohrt et al., 2019).".

530         Please note that N deposition is significantly underestimated due to canopy exchange processes as discussed by Bobbink et al., 1992 (in Environmental Pollution), Talkner et al. 2010 (in Plant and Soil) and others.

**Comment author**: Thank you for this comment. Assuming that N depositions are even higher than reported due the mentioned underestimation, would strengthen our argumentation, as the difference between P fluxes relative to reported P deposition and N fluxes relative to reported N depositions would be even larger.

The revised sentence in lines 480ff reads as follows:

"In comparison, N fluxes from the A horizons are only 30% and 28% of atmospheric N depositions that have been measured at the same high-P and the low-P site (Brumme et al., 2021; NW-FVA, 2020), which might even represent an underestimate as atmospheric N inputs are generally not completely captured due to canopy exchange processes (Talkner et al., 2010)."

---

## Author Comment (AC2)

Response to Reviews of: **Leaching of inorganic and organic phosphorus and nitrogen in contrasting beech forest soils – seasonal patterns and effects of fertilization**

**Review comment #2**

Fetzer an co-authors studied element fluxes woth percollating water through upland soil profiles. Their work focuses on P fluxes. They compare (a) two sites (high/low P) (b) three different depths (litter, organic layer, A horizon), (c) seasonal dyanmics, and (d) the effects of N, P, and N+P fertilization. The authors aimed for a semi-experimental approach, where heavy rainfall event are simulated at each site to measure soil leachate concentrations under comparable rainfall conditions. Their key findings are that (a) season is the most important determinant of P fluxes, (b) inorganic N and P shows stronger sesasonal variation than organic P fluxes (c) there were surpsiningly small differences in P fluxes between the two sites, but the two sites responded differently to fertilization, in paticulary N+P treatments.

**Strength:**

This is a timely study addressing a important topic - P dynamics in soil profiles less well understood than C and N dynamcis. The authors used state-of-the-arts methods and their results justify their conclusions. Overall, this is an impressive piece of work that features a fully factorial experiment with 5 independent variables (site, horizon, season, +N, +P) and over 10 measured endpoints (concentrations and fluxes of DIP, DOP, DON, DIN, DOC).

**Comment author**: Thank you for this positive feedback.

**Weaknesses:**

1. I think the scope of the expriment is also a main limitation to the manuscript. I cannot get rid of the feeling that the authors tried to do too much in one step here. This has some consequence in experimental design: The authors tried to study both 'background' (unfertilized) fluxes and fertlization effectes at the same time. This made compromises in experimental design necessary like the application of KCl to control plots to compensate for the applied K in P fertilization plots. This raises the question how representative the control fluxes still are for natural conditions.

**Comment author**: Thank you for the comment. While writing the manuscript, we also discussed intensively what to include in the manuscript or not. We opted on presenting and discussing also the leaching of the control plots to present a baseline and the relevance of P leaching as this information is rather scarce (as noted by the reviewer). Moreover, the discussion of fluxes in the control plots, documenting that estimated P fluxes at our sites correspond to those obtained by other leaching studies at the same site and elsewhere is needed to interpret the fertilization experiment.

We think that it is unlikely that the KCl addition affected P leaching as chloride is less competitive in sorption than inorganic and organic P forms. Indirect effects on sorption/desorption via changes in ionic strength seem unlikely as the measured electrical conductivity was $63 \pm 45$ µS cm$^{-1}$ (average $\pm$ St. dev for all samples) which in the typical range of soil solutions sampled in forest topsoils. Therefore, we do not expect increased P desorption and fluxes by the KCl addition in organic layers and A horizons. Our assumption is supported by comparable P concentrations and fluxes from our control plots (where KCl was added) to measured P concentrations and fluxes by other groups at the same sites (unpublished, values see responses to (3) and at line 445) and elsewhere (e.g. Sohrt et al., 2019).

2. I think the size and complexity of the presented project also limited the degree to which individual results are discussed. Overall, the discussion section remains largely limited to providing explanations for the observed phenomena. I think this undersells the novelty and significance of the presented data. It would be nice to hear not only how the observations can be explained, but also how they changed your conceptual understanding of the soil P cycle? What are the implications of your findings?

**Comment author**: We agree with the reviewer that the broad scope of this study is the advantage and the weakness. Although some findings might be undersold, we opted for presenting a comprehensive view to P cycling in forest soils and think that our study clearly shows so far rarely considered aspects such as the combination of N and P status, seasons, and environmental conditions. We think the manuscript's true novelty is to bring all these factors together instead of slicing the manuscript.

Only this combination allowed us to draw the conclusions that (1): the cycling of P and N may undergo considerable decoupling (indications from comparison of sites as well as fertilization treatments) and (2) that nutrient-poor ecosystems that recycle their nutrients tend to be vulnerable to changes in environmental conditions, such as seasonality, drying-rewetting, as well as external nutrient inputs. These are important contributions to the understanding of the soil P cycle in forests that were discernable only by a complex experimental design as used here.

3. I think the experimental approach chosen (field measurements but with the same rain event simulation performed at both field sites) and the consequences of these choices need to be discussed more explicitly. How representative are these simulated heavy rain events for 'normal' conditions with much smaller rainfall event spread out over the year? What did you learn about this new experimental approach?

**Comment author**: Thank you for this comment, this is a fair point. In the revised manuscript, we discussed this more in depth and this is the reason why we added the information about rainfall intensities and annual precipitation in the Methods section (lines 143ff):

"The application rate represents maximum rainfall intensities at the study sites. Rainfall intensities larger than 20 L h$^{-1}$ m$^{-2}$ have been observed once at the low-P site and three times at the high-P site during the last 10 years (Bayerische Landesanstalt für Wald und Forstwirtschaft (LWF) and Nordwestdeutsche Forstliche Versuchsanstalt (NW-FVA)). **The amount of water added with irrigation corresponds to the average weekly precipitation at the high-P site and exceeds it by 33% at the low-P site**. In 2018, the three irrigations, totaled 60 L m$^{-2}$, which accounted for approx. 8% of measured throughfall at the high-P site and 16% at the low-P site (cf. Table 2). The two irrigations in 2019 added 40 L m$^{-2}$."

Additionally, we added mean annual precipitation data in the description of the sites in lines 94 and 100.

These additional information shows that the total addition of artificial rainwater was little compared to annual precipitation, and therefore, did not change strongly the annual fluxes and falls within the amounts of weekly rainfalls.

We compared our data with unpublished P fluxes under ambient conditions from the organic horizons at the same sites during the previous four years. Our fluxes were slightly smaller, which could be due to less precipitation in the studied year than in the previous four years. Therefore, we are confident that our flux estimations based on P concentrations obtained by artificial irrigation are reliable and representative for natural conditions. In the revised manuscript, we provide the comparison to these data in the Discussion and discuss the representativeness as follows (lines 445ff):

"Dissolved P concentrations in the leachates following the experimental irrigation used to overcome site and weather variations corresponded closely to those measured in an adjacent plot receiving natural precipitation. While the annual average concentration in the leachate from the organic layer (only control plots) following irrigation were 0.19 mg P L$^{-1}$ at the low-P site and 0.24 mg P L$^{-1}$ at the high-P site, respectively, those under natural precipitation were 0.35 mg P L$^{-1}$ at the low-P site and 0.18 mg P L$^{-1}$ at the high-P site (K. Kaiser, unpublished data, median over the four previous, much wetter years). We therefore assume that concentrations and fluxes estimated here, are representative for the sites. The TDP fluxes, ranged between 12 and 60 mg total P m$^{-2}$ yr$^{-1}$ across all horizons (Table 3), compare well with the P fluxes measured in other forest ecosystems, ranging from 9 to 62 mg P m$^{-2}$ yr$^{-1}$ (Qualls, 2000; Fitzhugh et al., 2001; Hedin et al., 2003; Piirainen et al., 2007; Sohrt et al., 2019; Rinderer et al., 2021)."

4. Finally, it's not quite clear to me how the annual fluxes were calcualted. I'm assuming that these were upascaled from the concentrations found from the soil leaching experiments perfromed 4x/year? If that's true, I would doubt that the concentrations measured in such experiments are representative for other (less intense) rain events throughout the year. I would also assume that leachate P concentrations vary with the length/intensity of individual rain events, and the length of and conditions during the periods between rain events. All in all, I'm not convinced that the presented data allows calcualting and annual P balance that can be compared in absolute terms (e.g. to deposition inputs).

**Comment author**:

**Methods**

Correct, we upscaled the concentrations from the point measurements and multiplied them with measured water fluxes. In the revised manuscript, we rephrase and expand the describing of the approach used for flux estimation in lines 218ff. We are aware that these flux estimates are approximations (as in many other studies where measured element concentrations are multiplied with modelled water fluxes). A continuous monitoring at the 2 sites receiving NxP fertilizer would not have been possible. Nonetheless, we regard them to correspond to other assessments (see last response and Discussion lines 445ff). In the Discussion, we present the number very cautious and present the numbers in a rather conservative manner ("ranged between 12 and 60 mg total P m$^{-2}$ yr$^{-1}$ across all horizons (Table 3)"; "The P fluxes from the A horizon at the high-P and at the low-P site are approx. 150% and 50% of reported atmospheric P deposition in Germany." Therefore, we are confident that we are sufficiently cautious in our data interpretation.

Reply to the Reviewers

**How representative are the concentrations obtained by artificial irrigation with 20 L m$^{-2}$ h$^{-1}$ for other (less intense) rain events?**

**Firstly**, the amount of water added with irrigation corresponds to the average weekly precipitation at the high-P site and exceeds it by 33% at the low-P site (added at line. 145). We regard this as an amount representative of a higher intensity rainfall event. **Secondly**, in order to have enough organic layer leachate reaching the mineral horizon, also a certain amount of precipitation is needed, especially on dry soils. The amount of artificial rainfall we applied correspond to 60-70% of the pore volume of the soil material above the lysimeter in the A horizon (approx. 1 L of water for the area of a lysimeter (19.5 * 25.5 cm). We therefore think, the applied amount was an appropriate comprise between "representative" conditions and the need to obtain sufficient leachate for analysis. **Thirdly**, P concentrations indeed vary with length of rain events (see reference in lines 142ff: Our sampling procedure represents the "first flush", comprising the majority of P leached during heavy rainfall events (Bol et al., 2016; Makowski et al., 2020a; Rinderer et al., 2020). In terms of length of rain events, there is a decrease in P towards the end due to dilution when P concentrations reach a constant low level (Rinderer et al., 2020). Therefore, most P export happens during the "first flush", which we covered by our experiment. The close match of P concentrations measured here and in the continuous monitoring supports our assumption that we have sampled representative leachates. As mentioned above, this information has been added to the manuscript.

Also, please note that the standardized irrigation allowed a better comparison between sites, treatments, and seasons.

Our annual fluxes are clearly estimates. Therefore, we compared our concentrations and fluxes to other studies that obtained their data under natural rainfall conditions (see comment above). As the concentrations and fluxes were similar, we are confident that our data is a sound approximation of natural conditions and we think it is useful to set it in comparison with other numbers (that are often estimates as well since being based on modelled water fluxes), to judge the importance of the fluxes.

**Possibilities for improvement:**
1. I would suggest adding some graphic summary of the main findings (e.g. a conceptual figure).

2. I would suggest removign part of the data. Alternatively (in my opinion, preferably) would be splitting the mansucript into two companion papers (e.g., one dealing with site, horizon, and season; the second with fertilization effects). This would give more space to discuss the novelty and implications of each part of the study.

**Comment author**: We appreciate your suggestions and constructive thoughts on the manuscript. While writing the manuscript, we also considered splitting, but opted on providing a more holistic assessment of complex ecological interactions under field conditions. We felt that for the evaluation of the fertilization effect, we first have to document and discuss the representativeness of the measured fluxes (varying differently for DIP and DOP at a seasonal scale). Moreover, fertilization effects depended upon sites and therefore, we have to discuss 'site effects' beforehand.

In conclusion, we prefer to keep the manuscript as one.

Due to the complexity of the variables, factors and processes involved (DIP, DOP, DIN, DOP, 3 horizons, 2 sites, interaction of N x P fertilization), we also refrained from providing a conceptual

figure, which would be too simplistic. We could instead provide a kind of summary graph as this one:

[Figure]

Additionally, we tried our best to revise the result and discussion section to improve the clarity of the processes involved and provide a deeper insight into P cycling.

**Minor comments:**
I would avoid using the term climate to refer to seasonal dynamics (eg. L518).

**Comment author**: Thank you for spotting this inconsistency. We changed the term to seasonal conditions (line 552).